# Detecting and Pruning Prominent but Detrimental Neurons in Large Language Models

**Ameen Ali, Shahar Katz and Lior Wolf**
The Blavatnik School of Computer Science
Tel Aviv University
{ameenali@mail,shaharkatz3@mail,wolf@cs}.tau.ac.il

**Ivan Titov**
The University of Edinburgh
The University of Amsterdam
ititov@inf.ed.ac.uk

## Abstract

Large language models (LLMs) often develop learned mechanisms specialized to specific datasets, such as reliance on domain-specific correlations, which yield high-confidence predictions without generalizable reasoning. While beneficial in one setting, these dataset-specific mechanisms typically degrade performance when models encounter novel tasks or distributions. In this work, we introduce a fine-tuning approach designed to enhance generalization by identifying and pruning neurons associated with dataset-specific mechanisms in transformer-based LLMs. Our method employs Integrated Gradients to quantify each neuron's influence on high-confidence predictions, pinpointing those that disproportionately contribute to dataset-specific performance without supporting robust, transferable reasoning. Selectively pruning these neurons compels the model to depend on generalizable representations. Evaluated across multiple-choice benchmarks, our pruning-based fine-tuning significantly enhances performance, surpassing prior (non-pruning) adaptation methods.

## 1 Introduction

Large language models (LLMs) have demonstrated remarkable performance across various natural language tasks, including question answering, reasoning, and knowledge retrieval. However, recent studies have revealed a concerning pattern: these models often achieve high performance by exploiting spurious correlations in the training data rather than through genuine understanding of the underlying tasks—a phenomenon commonly referred to as "shortcut learning" (Yuan et al., 2024). This manifests when models leverage superficial patterns that correlate with target outputs in the training distribution but fail to capture the causal mechanisms or deeper semantic relationships necessary for reliable generalization to out-of-distribution scenarios. For instance, a model might associate specific keywords with particular answer choices in multiple-choice questions rather than developing a comprehensive understanding of the problem, leading to significant performance degradation when these spurious correlations break down in real-world applications. Previous approaches to mitigating shortcut learning have focused on data-centric solutions, such as augmenting training data with counterfactual examples (Feder et al., 2023), adversarial training (Altinisik et al., 2022), or ensemble methods that combine multiple objectives (Jiang et al., 2023b).

The term shortcut implies that the model reaches a decision more quickly, i.e., at earlier layers. It also suggests reliance on heuristics that may lack a robust foundation. However, decision mechanisms encoded as neural pathways can become irrelevant when shifting to new distributions, regardless of their speed or broad applicability within the original training distribution.

Our work focuses on the elimination of such domain-specific mechanisms (DSM) as part of a fine-tuning process, specifically in the context of multiple-choice tasks. This type of task allows us to directly identify mechanisms that may have been beneficial in the original domain but become detrimental in the target domain by relying on the prediction accuracy.

Our method is based on the following observations: (i) to eliminate a DSM, it is sufficient to prune certain neurons within the relevant pathway; (ii) inference mechanisms contributing to the decision-making process can be identified using explainability methods such as Integrated Gradients (Sundararajan et al., 2017); and (iii) eliminating inference mechanisms harmful in the target domain would lead to improved accuracy in that domain.

Based on the first observation, the proposed fine-tuning method prunes specific neurons within a single layer, which, as we demonstrate, is sufficient for effective fine-tuning. Drawing on the second observation, candidate neurons are identified using IG as those most prominent during inference on a small set of fine-tuning samples. This makes our method suitable for few-shot fine-tuning. It is also noteworthy that this stage does not require any label information. The number of neurons pruned and the specific layer are determined based on the accuracy achieved on the same sample set.

One key advantage of our approach is its efficiency in fine-tuning. Unlike full model finetuning strategies, our method operates on a minimal set of up to 10 samples to identify DSM neurons. Using this same set, we determine both the optimal layer for pruning and the percentage of neurons to remove. This makes our method highly practical for scenarios where labeled data is scarce or costly to obtain.

Additionally, because pruning is applied selectively within a single layer, the cost remains significantly lower compared to full fine-tuning or re-training approaches. Despite its simplicity, our method yields substantial improvements in multiple-choice benchmarks. As can been seen from Figure 1, our method improves results across three model architectures: LLaMA 3.1-8B Instruct, Mistral-7B-v0.3 Instruct, and Qwen2.5-7B Instruct.

## 2  Related Works

Research on improving generalization in large language models (LLMs) spans several interconnected areas. This section reviews prior work in (1) shortcut learning in neural networks, (2) neuron-level interpretability in LLMs, (3) attribution methods for deep networks, and (4) model pruning approaches.

**Shortcut Learning in Neural Networks**   Shortcuts refer to the phenomenon whereby models generate responses using shallow heuristics rather than engaging in a deep semantic understanding of a given task. LLMs are known to frequently rely on such shortcuts (Du et al., 2023; Ho et al., 2022; Levy et al., 2023), allowing them to address various reasoning tasks by exploiting superficial patterns, such as common n-grams. However, this reliance may prevent the models from fully capitalizing on their inherent reasoning capabilities, ultimately resulting in suboptimal performance. Notably, Du et al. (2021) employed Integrated Gradients (Sundararajan et al., 2017) to identify input tokens that function as shortcuts in encoder-only models such as BERT (Devlin et al., 2019). Subsequent studies have extended this line of research by utilizing saliency-based gradient methods and token statistics to detect shortcuts either in model inputs or within training datasets (Bastings et al., 2022; Zhao et al., 2024; Friedman et al., 2022; Tang et al., 2023), proposing enhanced training strategies to mitigate shortcut reliance. In this work, we investigate whether pruning a small subset of LLM parameters can reveal latent reasoning abilities that are otherwise suppressed by shortcut-related parameters.

**Neuron-level Interpretability in LLMs**   Research on neuron-level interpretability in LLMs has evolved to focus on the detailed roles played by individual neurons. Early studies revealed that some neurons store specific types of information. For instance, the Knowledge Neurons hypothsis (Dai et al., 2022) showed that particular neurons can hold factual details, prompting further investigation into how such localized information influences model behavior. Two main approaches have been employed to study neuron function: activation analysis and attribution methods. Activation analysis examines the responses of neurons to a variety of inputs, uncovering patterns that correlate with features such as syntax, semantics, recurring n-grams and confidence (Voita et al., 2023; Gurnee et al., 2024; Stolfo et al., 2024). Attribution methods, like Integrated Gradients (Sundararajan et al.,

2017), estimate the contribution of individual neurons to the final output. Together, these techniques allow researchers to classify neurons into functional groups and identify those that are critical for specific behaviors. Moreover, recent work has applied these methods to multilingual models (Tang et al., 2024), distinguishing between neurons that operate in a language-specific manner and those that are language-agnostic, Ali et al. (2024) proposed a modification to the Integrated Gradient method to allow the discovery of copying neurons under the In-Context learning settings.

**Model Pruning and Parameter-Efficient Fine-tuning**  LLMs are typically trained for general-purpose applications, enabling them to handle a wide range of tasks in a zero-shot setting. To enhance performance on specific tasks, a common approach is to continue training the models on targeted samples - a process known as fine-tuning (Houlsby et al., 2019; Brown et al., 2020; Han et al., 2024). However, fine-tuning is computationally expensive and demands more than just a handful of examples. To mitigate the computationally challenge, more efficient strategies such as LoRA (Hu et al., 2022) have been developed, which focus on modifying only a small subset of the model's parameters to reduce computational overhead.

However, fine-tuning, as well as fine-tuning with reinforcement learning from human feedback (RLHF) (Ouyang et al., 2022), requires extensive annotated data, which are not always available. Addressing this limitation, recent studies propose methods that adapt models using only a few training samples. One notable example is Inference-Time Intervention (ITI) (Li et al., 2023a), which enhances performance on QA benchmarks by steering the activations in the attention modules using merely 40 annotated samples to map the geometric direction associated with truthfulness. Similarly, SADI (Wang et al., 2025) employs a steering mechanism on both model activations and the neurons within the MLPs' layers with just 150 samples.

In this work, we demonstrate that disabling a small subset of parameters—by zeroing them out rather than retraining—can outperform previous state-of-the-art methods using fewer samples and without relying on annotated labels.

## 3   Method

In this section, we describe our novel approach for enhancing multiple-choice performance in transformer-based large language models (LLMs) via targeted neuron pruning. Our method is motivated by three key ideas. First, we analyze the role of multi-layer perceptrons (MLPs) within transformer architectures, which are central to encoding and processing the bulk of an LLM's knowledge. Second, we build upon the concept of *knowledge neurons*—specialized units that store factual and task-relevant information—to motivate our extension. Finally, we leverage Integrated Gradients (IG), originally designed for input feature attribution, to identify neurons that drive high-confidence yet domain specific mechanisms (DSM neurons), and we prune them to promote more robust reasoning.

### 3.1   MLPs in Transformer LLMs and Knowledge Neurons

Transformer models are composed of a series of blocks, each typically featuring a multi-head self-attention mechanism followed by a feed-forward network. The MLP sub-layer is responsible for non-linear transformations that significantly shape the model's internal representations. Formally, for the $l$-th layer, the MLP can be expressed as:

$$\text{MLP}(x_i^l) = \sigma(W_1^l x_i^l + b_1^l)W_2^l + b_2^l, \tag{1}$$

where $x_i^l \in \mathbb{R}^d$ is the hidden state of the $i$-th token, $W_1^l \in \mathbb{R}^{d_m \times d}$ and $W_2^l \in \mathbb{R}^{d \times d_m}$ are the weight matrices, $b_1^l$ and $b_2^l$ are biases, and $\sigma(\cdot)$ is a non-linear activation function (e.g., ReLU or GELU). Recent research in transformer-based language models has revealed that not all neurons contribute uniformly to a model's performance. A subset of neurons, termed *knowledge neurons*, have been shown to encode and store specific factual or task-related information. These neurons often exhibit strong activations in response to inputs that

invoke particular pieces of knowledge, indicating their role in recalling and processing this information. The identification of knowledge neurons has not only deepened our understanding of the internal workings of LLMs but also paved the way for targeted interventions, such as controlled model editing and selective pruning, to enhance model robustness and generalization.

## 3.2 Integrated Gradients for Attribution

Integrated Gradients (IG) is a principled attribution method developed to explain the predictions of deep neural networks. It addresses limitations of naïve gradient-based techniques, such as gradient saturation, by averaging the gradients along a straight-line path from a baseline input to the actual input.

Formally, given a model $f$ and an input $x$, the attribution for the $i$-th feature is defined as:

$$\text{IG}_i(x) = (x_i - x_i') \int_0^1 \frac{\partial f(x' + \alpha(x - x'))}{\partial x_i} \, d\alpha, \tag{2}$$

where $x'$ is a baseline input that represents the absence of features (e.g., a zero vector) and $\alpha$ scales the interpolation between the baseline and $x$. The method satisfies key axioms such as *sensitivity*—ensuring that features affecting the output receive non-zero attributions—and *implementation invariance*—which guarantees that functionally equivalent models yield identical attributions.

In practice, the integral is approximated using a Riemann sum over $m$ discrete steps:

$$\text{IG}_i(x) \approx (x_i - x_i')\frac{1}{m} \sum_{k=1}^{m} \frac{\partial f\left(x' + \frac{k}{m}(x - x')\right)}{\partial x_i}. \tag{3}$$

Due to its solid theoretical foundations and empirical success, Integrated Gradients has become a widely adopted tool for model interpretability in areas such as computer vision and natural language processing. Although IG was formulated for input features, its conceptual framework can be extended to internal neurons. In our context, we focus on computing IG with respect to the token with the highest logit of the model's output distribution—which encapsulates the model's most confident prediction. This extension allows us to quantify the influence of individual neurons on the model's decision. In particular, the decision to calculate IG with respect to the maximum logit, ignores the actual annotated labels of given sample.

## 3.3 DSM Neuron Detection and Pruning

Building on the ideas of knowledge neurons and Integrated Gradients, we hypothesize that a subset of neurons in the MLPs act as DSM neurons that disproportionately drive the maximum logit by exploiting spurious correlations rather than robust reasoning. Instead of interpolating over the input, we interpolate over the weight of the neuron under consideration. For a given question $q$ (e.g., from MMLU, SST, or BoolQ) and its associated output logits $f(q)$, let

$$M(q) = \max_{y \in V} f(q)_y,$$

where $V$ is vocabulary space.

Let $\hat{w}_j$ denote the original weight of neuron $n_j$ in the MLP. We define the Integrated Gradients (IG) attribution for neuron $n_j$ by interpolating its weight from a baseline value of 0 to $\hat{w}_j$. Formally, the attribution is given by

$$\text{IG}(n_j, q) = \hat{w}_j \int_0^1 \frac{\partial M(q; \alpha \hat{w}_j)}{\partial w_j} \, d\alpha, \tag{4}$$

where $M(q; \alpha \hat{w}_j)$ represents the output of the original LLM (e.g., the logit corresponding to the model's prediction) when the weight of neuron $n_j$ is scaled by $\alpha$. The term $\frac{\partial M(q; \alpha \hat{w}_j)}{\partial w_j}$

captures the sensitivity of the model's output to the modified neuron weight at interpolation step $\alpha$. In practice, we approximate the integral via a Riemann sum over $m$ steps.

We then aggregate the IG attributions over a dataset $\mathcal{D} = \{q_i\}_{i=1}^N$ to obtain an average attribution score for neuron $n_j$:

$$\text{score}(n_j) = \frac{1}{N} \sum_{i=1}^N \text{IG}(n_j, q_i). \tag{5}$$

Calculating IG with respect to the maximum logit, ignores the label of the given sample.

### 3.4 Mitigating DSM Neurons

After computing the IG attribution scores for all neurons and averaging over the validation set, we obtain a score matrix of shape $L \times d$, where $L$ is the number of layers and $d$ is the number of neurons per layer. These scores indicate the relative importance of each neuron in driving the model's predictions.

Since these attributions are derived from the model's output, pruning highly attributed neurons at random may degrade performance. However, in some layers, these neurons contribute to spurious correlations rather than meaningful reasoning. To identify such layers, we apply a grid search over the layers and pruning percentages. Specifically, we iteratively prune the top-attributed neurons per layer in increments of 5%, ranging from 5% to 40%, and evaluate the model's performance on a held-out validation set.

This systematic search allows us to determine which layers benefit from pruning, leading to improved generalization by reducing reliance on spurious correlations, while also identifying layers where pruning degrades performance, indicating that their high-attributed neurons are essential for robust reasoning. This targeted pruning strategy helps refine the model's decision-making without arbitrary neuron removal. The exact psuedo-code for our algorthim is presented in Algorthim 1.

---

**Algorithm 1** DSM Neuron Pruning

---

**Require:** Task-specific dataset $\mathcal{D} = \{q_i\}_{i=1}^N$, LLM $\mathcal{M}$, validation set $\mathcal{D}_{\text{val}}$, $L$ number of layers in $\mathcal{M}$.
**Ensure:** Task-optimized pruned model $\mathcal{M}_{\text{pruned}}$
1: **for** each question $q \in \mathcal{D}$ **do**
2:     Compute $M(q) = \max_{y \in V} f(q)_y$                             ▷ Maximum logit
3:     **for** each neuron $n_j$ in model $\mathcal{M}$ **do**
4:         $\text{IG}(n_j, q) = \hat{w}_j \int_0^1 \frac{\partial M(q; \alpha \hat{w}_j)}{\partial w_j} d\alpha$
5:         $\text{score}(n_j) += \text{IG}(n_j, q)/N$                ▷ Aggregate attribution
6:     **end for**
7: **end for**
8: $(l_{\text{opt}}, p_{\text{opt}}) \leftarrow \underset{l \in [1, L], p \in \mathcal{P}}{\arg\max} \text{Eval}(\text{Prune}(\mathcal{M}, l, p), \mathcal{D}_{\text{val}})$     ▷ $\mathcal{P} = \{5\%, ..., 50\%\}$
9: $\mathcal{M}_{\text{pruned}} \leftarrow \text{Prune}(\mathcal{M}, l_{\text{opt}}, p_{\text{opt}})$
10: **return** $\mathcal{M}_{\text{pruned}}$

---

## 4 Experiments

In this section, we evaluate the effectiveness of our proposed method for detecting and pruning DSM neurons in transformer-based large language models (LLMs). Our primary goal is to assess whether selectively pruning neurons identified as DSM neurons can improve model generalization and performance across a variety of natural language tasks. To do so, we conduct a series of experiments on benchmark datasets commonly used in multiple-choice and reasoning tasks.

**Tasks**    Our experiments focus on multiple-choice tasks. We evaluate performance using the following datasets: XNLI (Bowman et al., 2015), MMLU (Hendrycks et al., 2020), SST2 (Socher et al., 2013), SST5 (Socher et al., 2013), BoolQ (Clark et al., 2019), and Balanced COPA (Roemmele et al., 2011). These datasets include response formats ranging from two to five choices. A detailed description of each dataset is provided in Appendix A. Accuracy serves as the primary evaluation metric.

**Models**    We evaluate our method on a diverse set of instruction-tuned LLMs, including LLAMA2-7B-CHAT[1] (Touvron et al., 2023), LLAMA3.1-8B-Instruct[2] (Grattafiori et al., 2024), Mistral-7B-Instruct[3]-v0.3 (Jiang et al., 2023a) and Qwen2.5-7B-Instruct[4] (Jiang et al., 2023a). We use these models due to their strong performance on various NLP benchmarks and their widespread adoption in research. By including models with different training methodologies and parameter sizes, we aim to assess the robustness and adaptability of our proposed method across multiple instruction-following paradigms.

**Baselines**    We compare our few-shot approach against several established adaptation methods. Supervised Fine-Tuning (SFT) serves as an upper bound for supervised adaptation by fine-tuning all model parameters using the complete training dataset. We also compare against the following adaptation methods: Inference-Time Intervention (ITI) (Li et al., 2023a) leverages contrastive pairs to identify key attention heads for intervention, with a hyperparameter sweep over head selection and intervention strength. This method typically requires dozens of annotated examples. Contrastive Activation Addition (CAA) (Rimsky et al., 2024) constructs a fixed steering vector by computing the mean activation difference at the answer position between positive and negative prompts, and generally uses substantial training data. SADI (Wang et al., 2025) is evaluated in three configurations: SADI-HIDDEN, which applies SADI to key hidden states across the layers; SADI-HEAD, which modifies activations from all attention heads across layers; and SADI-NEURON, which adjusts outputs from non-linear activation functions in the FFN blocks. These methods require access to many annotated training samples. In contrast to these approaches, our DSM neuron pruning method operates in a genuine few-shot setting, employs only a handful of examples (ten in our experiments) to identify DSM neurons and determine optimal pruning configurations. This makes our method particularly valuable when labeled data is scarce or expensive to obtain.

**Implementation Details**    In all our experiments, we apply the Integrated Gradients (IG) method to the gate_proj layer within each MLP block of the transformer. The shape of this layer varies across architectures, as detailed in Appendix B, the exact promoting format is illustrated in Appendix C . The samples used within the Integrated Gradients framework are picked from the training split in a random way, while the reported accuracy is reported over the test sets.

## 5    Results

### 5.1    Quantitative Experiments

We evaluate our DSM neuron pruning method across multiple tasks and model architectures. Tables 1-3 summarize the performance of our method compared to baseline models and other adaptation approaches. Table 1 presents results for LLAMA2-7B-CHAT across five multiple-choice tasks. Our method achieves significant improvements over the baseline model, with absolute gains of 0.67%, 4.78%, 4.91%, 7.60%, and 3.6% on NLI, MMLU, SST2, BoolQ, and Balanced COPA respectively. Notably, our approach outperforms other parameter-efficient adaptation methods including ITI (Li et al., 2023b), CAA (Rimsky et al., 2024), and various SADI variants. On MMLU, our method achieves 49.68% accuracy compared to the baseline's 44.90%, representing a substantial improvement on this challenging

---

[1] https://huggingface.co/meta-llama/Llama-2-7b-chat-hf
[2] https://huggingface.co/meta-llama/Llama-3.1-8B-Instruct
[3] https://huggingface.co/mistralai/Mistral-7B-Instruct-v0.3
[4] https://huggingface.co/Qwen/Qwen2.5-7B-Instruct

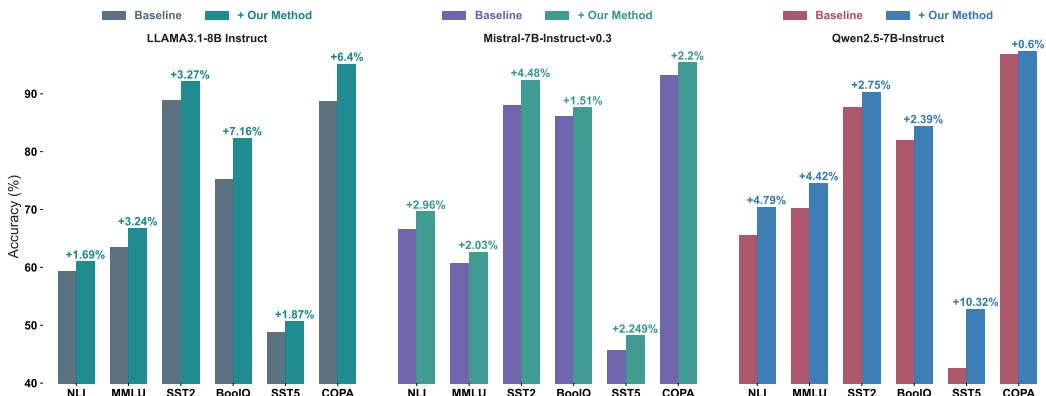

Figure 1: Comparative analysis of performance improvements across three LLM architectures using our method. Each model family (LLAMA3.1-8B Instruct, Mistral-7B-v0.3 Instruct, and Qwen2.5-7B Instruct) shows consistent accuracy gains across six standard NLP benchmarks.

benchmark. For sentiment analysis (SST2), our approach reaches 93.54% accuracy, approaching the performance of supervised fine-tuning (96.70%) without requiring labeled data. The random pruning baseline generally fails to improve performance over the original model, confirming that our targeted identification of DSM neurons is essential rather than simply reducing model parameters. Table 2 shows the effectiveness of our method on the

| Task | NLI | MMLU | SST2 | BoolQ | Balanced COPA |
|------|-----|------|------|-------|---------------|
| LLAMA2-7B-CHAT | 63.11 | 44.90 | 88.63 | 70.52 | 70.80 |
| + ITI | 63.97 | 46.07 | 91.38 | 74.10 | - |
| + CAA | 64.13 | 46.17 | 91.16 | 74.98 | - |
| + SADI-HIDDEN | 59.28 | 45.66* | 92.15 | 76.25 | - |
| + SADI-NEURON | 62.97 | 46.91* | 88.69 | 70.40 | - |
| + SADI-HEAD | **64.21** | 48.23* | 92.20 | 74.35 | - |
| + Random Pruning | 63.22 | 44.10 | 88.65 | 70.51 | 66.00 |
| + Ours | 63.78 | **49.68** | **93.54** | **78.12** | **74.40** |
| + Supervised Finetuning | 90.07 | - | 96.70 | 88.75 | - |

Table 1: Overall performance of LLAMA2-7B-CHAT on the different 5 tasks in a zero-shot setting. *Uses transductive learning, i.e., all training samples are available upfront, only over the MMLU dataset.

LLAMA3.1-8B-Instruct, Mistral-7B-Instruct-v0.3, and Qwen2.5-7B-Instruct models. Consistent with our findings on LLAMA2, selective pruning of DSM neurons yields substantial gains across all benchmarks. For LLAMA3.1-8B-Instruct, the most notable improvements are observed on BoolQ (+7.16%) and SST2 (+3.27%), with COPA also showing significant improvement (+6.4%). Results on Mistral-7B-Instruct-v0.3 further validate the effectiveness of our method across different model families. Here, we observe improvements on all six tasks, with the most substantial gains on SST2 (+4.48%) and NLI (+2.96%). The Qwen2.5-7B-Instruct model shows the largest improvements in some categories, particularly SST5 (+10.32%) and NLI (+4.79%). The consistently positive results across Mistral, LLAMA3.1, and Qwen2.5 architectures demonstrate that DSM neurons are a common phenomenon across different transformer-based LLMs, and that our pruning method provides a broadly applicable solution. Importantly, random pruning shows virtually no improvement over the baseline models across all architectures, confirming that our targeted approach is identifying and removing specifically problematic neurons.

| Task | NLI | MMLU | SST2 | BoolQ | SST5 | Balanced COPA |
|------|-----|------|------|-------|------|---------------|
| LLAMA3.1-8B Instruct | 59.34 | 63.54 | 88.87 | 75.19 | 48.83 | 88.80 |
| + Random Pruning | 59.30 | 63.54 | 88.90 | 75.19 | 48.83 | 88.80 |
| + Ours | **61.03** | **66.78** | **92.14** | **82.35** | **50.70** | **95.20** |
| Mistral-7B-Instruct-v0.3 | 66.68 | 60.66 | 87.95 | 86.14 | 45.74 | 93.20 |
| + Random Pruning | 66.68 | 60.66 | 87.93 | 86.14 | 45.75 | 93.20 |
| + Ours | **69.64** | **62.69** | **92.43** | **87.65** | **48.23** | **95.40** |
| Qwen2.5-7B-Instruct | 65.66 | 70.17 | 87.61 | 82.04 | 42.53 | 96.80 |
| + Random Pruning | 65.66 | 70.20 | 87.61 | 82.04 | 42.53 | 96.81 |
| + Ours | **70.45** | **74.59** | **90.36** | **84.43** | **52.85** | **97.40** |

Table 2: Performance comparison of LLAMA3.1-8B Instruct, Mistral-7B-Instruct-v0.3, and Qwen2.5-7B-Instruct models across six NLP tasks (NLI, MMLU, SST2, BoolQ, SST5, and Balanced COPA), showing results for baseline models, random pruning, and our proposed method. Our method consistently outperforms both baseline and random pruning across all models and tasks.

While our primary evaluation focuses on English-language tasks, it is important to assess the multi-lingual generalization capabilities of our DSM neuron pruning approach. To this end, we extend our evaluation to multilingual scenarios, in which we apply DSM neuron pruning and evaluate on the same language without any change to the method or reliance on language-specific features or annotations. using the XCOPA [5] dataset (Ponti et al., 2020), which covers eight diverse languages: Indonesian (id), Italian (it), Swahili (sw), Tamil (ta), Thai (th), Turkish (tr), Vietnamese (vi), and Chinese (zh). Table 3 presents the performance comparison across these languages for LLAMA2-7B-CHAT, various SADI configurations, and our method. The results reveal interesting patterns about the effectiveness of different adaptation approaches in multilingual contexts. Our findings show that while SADI variants achieve the highest performance improvements in several languages, particularly Indonesian (id) and Italian (it), our method delivers consistent improvements across all languages without requiring transductive learning or labeled samples. For Swahili (sw) and Tamil (ta), our approach matches the performance of SADI variants, achieving the highest accuracy of 50.80% and 50.00%, respectively. It is worth noting that SADI-Neuron and SADI-Head achieve stronger performance on some languages, particularly Indonesian (id) with gains of 12.20% and 11.20%, respectively. However, these methods require transductive learning with all annotated samples from the training set, whereas our approach operates in a few shot manner with just 10 examples for the DSM neuron detection process. This dramatic reduction in data requirements (from hundreds of labeled samples to just 10) highlights the efficiency of our DSM neuron identification approach.

| Task | id | it | sw | ta | th | tr | vi | zh |
|------|-----|-----|-----|-----|-----|-----|-----|-----|
| LLAMA2-7B-CHAT | 51.40 | 61.20 | 50.20 | 49.40 | 50.80 | 49.40 | 51.80 | 62.80 |
| + SADI-Hidden* | 51.40 | 62.40 | 50.00 | **50.00** | 51.20 | 48.80 | 52.20 | 64.80 |
| + SADI-Neuron* | **63.60** | 68.80 | 50.20 | 48.80 | **53.80** | 50.60 | **60.40** | **70.40** |
| + SADI-Head* | 62.60 | **70.60** | **50.80** | 49.60 | 51.40 | **51.60** | 60.20 | 70.10 |
| + Ours | 55.40 | 67.40 | **50.80** | **50.00** | 50.80 | 51.40 | 54.80 | 68.60 |

Table 3: Evaluation results on multilingual task XCOPA with LLAMA2-7B-CHAT.

## 5.2 Ablation Studies

To assess the efficiency and robustness of our DSM neuron detection method, we conduct an ablation study on the number of samples required for Integrated Gradients (IG) computation. As computing IG attributions for all neurons in a model can be computationally expensive,

---

[5] https://huggingface.co/datasets/cambridgeltl/xcopa

determining the minimum number of samples needed for effective DSM neuron detection is crucial for practical applications.

Figure 2 presents the performance on BoolQ and SST2 tasks using the LLAMA2-7B-CHAT model with varying numbers of IG samples. The results demonstrate that our method achieves substantial improvements over the baseline even with very few samples. With just 5 samples, we observe improvements of 6.20% on BoolQ and 4.13% on SST2, highlighting the efficiency of our approach in identifying meaningful DSM neurons with minimal data. Interestingly, we observe that optimal performance is achieved with just 10 samples for both tasks, with accuracy values of 78.12% for BoolQ and 93.54% for SST2. Increasing the number of samples beyond this point does not yield consistent improvements and sometimes results in slightly lower performance. For instance, with 20 samples, performance drops slightly to 76.23% on BoolQ and 92.88% on SST2. Additionaly we examine the impact of transferring the optimal pruning configuration, derived from the adaptation set, to unseen test data, ensuring that the DSM neuron pruning method remains robust. Specifically, we compare the accuracy of the pruned model on both the test set and the adaptation set to assess the generalizability and effectiveness of the pruning configuration across different data splits, results can be found in Appendix D. Additional studies examining the robustness of our method to adaptation set selection, the effectiveness of single-layer versus multi-layer pruning strategies, sensitivity to pruning threshold hyperparameters, and the choice of intervention layer are provided in Appendix E. These additional results demonstrate the stability of our approach across various experimental configurations.

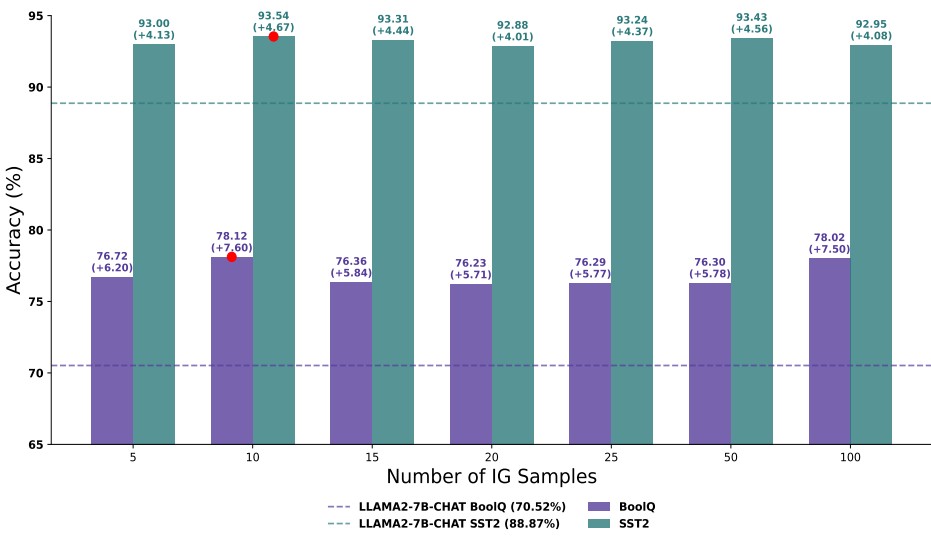

Figure 2: Ablation studies over the number of samples used within the IG framework to detect DSM neurons

## 5.3 Computational Cost Analysis

A practical consideration for our method is the computational cost of computing IG for all neurons in the model. To quantify this overhead, we conducted timing experiments across our four model architectures, measuring the time required to compute IG attributions for all neurons in the gate_proj layer. These experiments used 100 samples to obtain stable timing measurements and we report the average computation time per sample.

The results show that computing IG attributions requires between 3.74 and 4.68 seconds per sample across different models. Given that our method uses only 10 samples, the total computation time for DSM neuron detection ranges from approximately 37 to 47 seconds per task, a negligible overhead compared to traditional fine-tuning approaches. The dimensions

| Model | Avg. Time per Sample | Score Matrix Dimensions |
|---|---|---|
| LLAMA2-7B-CHAT | 4.38 sec | $32 \times 11{,}008$ |
| LLAMA3.1-8B-Instruct | 4.68 sec | $32 \times 14{,}336$ |
| Qwen2.5-7B-Instruct | 3.74 sec | $28 \times 18{,}944$ |
| Mistral-7B-Instruct-v0.3 | 4.46 sec | $32 \times 14{,}336$ |

Table 4: Computational cost of Integrated Gradients attribution and resulting score matrix dimensions for different models. The score matrix dimensions represent [number of layers $\times$ neurons per layer].

of the score matrix reveal the scale of our attribution analysis. For instance, LLAMA3.1-8B-Instruct produces a score matrix of shape $32 \times 14{,}336$, representing attribution scores for 14,336 neurons across each of the 32 layers, totaling 458,752 attribution values. Despite this large number of parameters being analyzed, our method efficiently identifies the most influential DSM neurons through ranking and selective pruning.

## 6 Conclusions

The idea that pruning improves performance is not new. In the first two years of life, the brain prunes excess synapses to improve neural efficiency Huttenlocher & Dabholkar (1997). This process is especially active in the visual and prefrontal cortices during key developmental windows Petanjek et al. (2011). In machine learning, pruning removes redundant or less important weights or nodes to reduce model complexity and overfitting LeCun et al. (1990). This can improve generalization and maintain or even enhance accuracy with fewer parameters Han et al. (2015).

What is unique about our work is that, unlike traditional pruning approaches which target redundant or low-importance parameters Sun et al., our method deliberately identifies and removes the most influential neurons—those driving high-confidence but non-transferable predictions. This inversion of the conventional pruning logic detects and addresses the 'empty vessels' (DSM) that 'make the most noise' (obtain the highest explainability scores), thereby suppressing mechanisms that impede generalization.

Moreover, our pruning is performed locally, within a single MLP layer, avoiding broad structural changes while still yielding substantial performance gains. We situate our approach within the broader context of transfer learning, specifically addressing the underexplored challenge of few-shot adaptation. Our results demonstrate that even with minimal data, selectively pruning influential neurons can significantly enhance cross-domain performance, offering a lightweight and interpretable alternative to conventional fine-tuning.

**Acknowledgements**

Ivan Titov is supported by the Dutch National Science Foundation (NWO Vici VI.C.212.053). This work was supported by a grant from the Tel Aviv University Center for AI and Data Science (TAD). This research was also supported by the Ministry of Innovation, Science & Technology ,Israel (1001576154) and the Michael J. Fox Foundation (MJFF-022407). The authors would like to thank Rochelle Choenni for the valuable insights. The contribution of Ameen Ali is part of a PhD thesis research conducted at Tel Aviv University.

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

# A Tasks

We evaluate our approach on a diverse set of benchmarks spanning commonsense reasoning, natural language inference, knowledge assessment, sentiment analysis, and question answering. In the following, we summarize the details and dataset sizes for each benchmark.

1. **Balanced COPA**[6]: A dataset where each question presents a premise and two alternatives, with the task of selecting the alternative that most plausibly has a causal relationship with the premise.

2. **XNLI**[7]: A multilingual extension of NLI, consisting of sentence pairs labeled as entailment, contradiction, or neutral, designed to evaluate cross-lingual natural language inference.

3. **MMLU**[8]: A benchmark assessing knowledge acquired during pretraining, covering 57 subjects across STEM, humanities, social sciences, and more.

4. **SST-2**[9] **and SST-5**[10]: Sentiment analysis datasets, with SST-2 providing binary labels (negative, positive) and SST-5 offering five labels (negative, somewhat negative, neutral, somewhat positive, positive).

5. **BoolQ**[11]: A question-answering dataset featuring naturally occurring yes/no questions.

| Task | Balanced COPA | XNLI (EN) | MMLU | SST2 | SST5 | Boolq |
|---|---|---|---|---|---|---|
| # train | 1K | 393K | 99.8K | 67.3K | 8.54k | 9.43k |
| # test | 500 | 5.01K | 14K | 872 | 2.21k | 3.27k |

Table 5: The number of data used for identifying key elements and testing for 6 tasks.

# B Intervention Layer

Our method is applied to the gate_proj layer across LLaMA, Mistral, and Qwen models. Specifically, we intervene in the transformation:

$$\text{MLP} = \text{down\_proj}(\text{act\_fn}(\text{gate\_proj}(x)) \times \text{up\_proj}(x)) \tag{6}$$

We choose the gate_proj layer as our operating layer for two key reasons. First, it plays a crucial role in the feedforward network (FFN) by modulating the interaction between the up_proj and down_proj transformations, making it a critical point of intervention. Second, this layer has a significantly larger parameter footprint than other FFN components, as detailed in Table 6, making it an effective target for structured modifications.

| Model | down_proj | up_proj | gate_proj |
|---|---|---|---|
| LLAMA2-7B-CHAT | $11008 \times 4096$ | $4096 \times 11008$ | $4096 \times 11008$ |
| LLAMA3.1-8B Instruct | $14336 \times 4096$ | $4096 \times 14336$ | $4096 \times 14336$ |
| Mistral-7B-Instruct-v0.3 | $14336 \times 4096$ | $4096 \times 14336$ | $4096 \times 14336$ |
| Qwen2.5-7B-Instruct | $152064 \times 3584$ | $3584 \times 18944$ | $3584 \times 18944$ |

Table 6: The number of data used for identifying key elements and testing for 6 tasks.

---

[6]https://huggingface.co/datasets/pkavumba/balanced-copa
[7]https://huggingface.co/datasets/facebook/xnli
[8]https://huggingface.co/datasets/cais/mmlu
[9]https://huggingface.co/datasets/stanfordnlp/sst2
[10]https://huggingface.co/datasets/SetFit/sst5
[11]https://huggingface.co/datasets/google/boolq

## C   Tasks Prompting

**SST2 Zero-Shot Prompt**

Consider the sentiment expression in this sentence and respond briefly with 'positive' or 'negative'.

{TEXT}

Answer:

**SST5 Zero-Shot Prompt**

Consider the sentiment expression in this sentence and respond briefly with 'very positive', 'positive', 'neutral', 'negative', and 'very negative'.

TEXT

Answer:

**MMLU Zero-Shot Prompt**

Question: QUESTION
Which of the following answers is correct?
A. CHOICES[0]
B. CHOICES[1]
C. CHOICES[2]
D. CHOICES[3]
State the letter corresponding to the correct answer.
Answer:

**XNLI Zero-Shot Prompt**

Answer whether the hypothesis is more likely to be true, false, or unclusive based on the given premise.
Premise: PREMISE
Hypothesis: HYPOTHESIS
Answer:

**Balanced COPA Zero-Shot Prompt**

Question: PREMISE Based on the previous passage, choose the most reasonable question.
A: CHOICES[0]
B: CHOICES[1]

Answer:

**BoolQ Zero-Shot Prompt**

Is the answer to the question encapsulated in the passage?   Please confirm

with 'yes' or 'no'.

Passage: PASSAGE

Question: QUESTION

Answer:

# D    Transferability from Adaptation to Test

To validate the robustness of our DSM neuron pruning method, we investigate whether the optimal pruning configuration identified on a small adaptation set transfers effectively to unseen test data. In our approach, we use the adaptation set not only to identify DSM neurons through Integrated Gradients, but also to determine the optimal layer and pruning percentage through a grid search. This dual-purpose use of the adaptation set raises an important question: does the configuration that performs best on the adaptation samples maintain its effectiveness when applied to entirely new examples?.

Figure 3 presents the results of our grid-search across the SST2 dataset using the LLAMA2-7B-CHAT model. We define two evaluation settings: "Test Pruned" represents the accuracy of the pruned model (at different layers) on the test set, where DSM neurons were detected using the adaptation set samples. "Adaptation Pruned" shows the accuracy of the pruned model on the same adaptation set samples that were used to detect the DSM neurons.

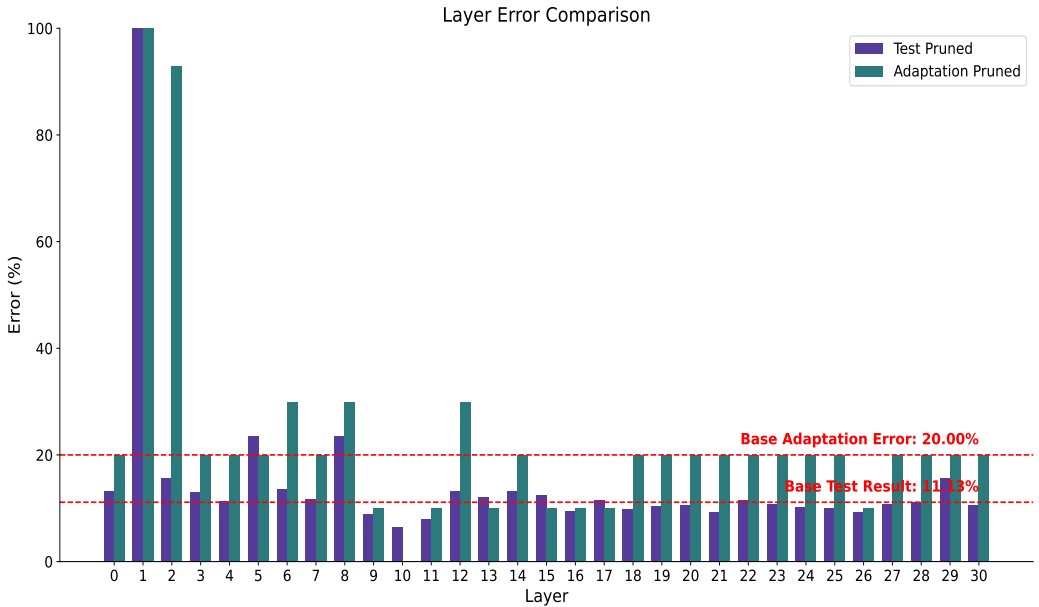

Figure 3: Comparison of error rates between pruned and non-pruned models across different layers on both adaptation and test sets for SST2

Figure 3 demonstrates that the optimal pruning configuration identified on the adaptation set effectively transfers to the test set. For the SST2 dataset, the non-pruned model achieves a baseline error rate of 20.0% on the adaptation set. By pruning layer 10, we achieve a perfect error rate of 0.0% on this same adaptation set. Importantly, this improvement transfers to the test set, where the pruned model (at layer 10) achieves 6.46% error rate (the lowest across the layers) compared to the non-pruned baseline of 11.13%.

# E    Additional Ablation and Robustness Studies

## E.1    Robustness to Adaptation Set Selection

A critical consideration for any few-shot adaptation method is its sensitivity to the specific samples chosen for the adaptation set. To evaluate the robustness of our DSM neuron pruning approach to this selection, we conducted comprehensive experiments examining both the variance across different random selections and the impact of sample quality.

**Variance Analysis Across Random Seeds** We performed five independent runs with LLAMA3.1-8B-Instruct on the BoolQ dataset, each using a different random seed to select the adaptation samples. Additionally, we compared the variance when using different adaptation set sizes (10, 50, and 100 samples) to understand how sample size affects stability.

| Adaptation Set Size | 10 Samples | 50 Samples | 100 Samples |
|---|---|---|---|
| BoolQ Accuracy | $82.93 \pm 0.65\%$ | $82.57 \pm 0.05\%$ | $82.43 \pm 0.02\%$ |

Table 7: Mean accuracy and standard deviation across five random seeds for different adaptation set sizes on BoolQ using LLAMA3.1-8B-Instruct.

The results demonstrate a high degree of stability in our method's performance. With just 10 samples, we achieve a mean accuracy of 82.93% with a standard deviation of only $\pm 0.65\%$, indicating that our DSM neuron identification is robust to the specific random selection. Interestingly, while larger adaptation sets (50 and 100 samples) show even lower variance, they do not improve mean performance, reinforcing our finding that 10 samples are sufficient for effective DSM neuron detection.

**Impact of Sample Correctness** To further investigate robustness, we examined whether the quality of adaptation samples, specifically whether the model correctly predicts them, affects the pruning effectiveness. We compared three scenarios: (1) randomly selected samples, (2) samples that the model correctly predicts, and (3) samples that the model incorrectly predicts.

| Dataset | Random | Correctly Predicted | Incorrectly Predicted |
|---|---|---|---|
| BoolQ | 82.35% | **85.25%** | 79.81% |
| SST2 | **92.14%** | **92.14%** | 87.50% |

Table 8: Performance comparison using different adaptation set selection strategies with LLAMA3.1-8B-Instruct.

These results reveal several important insights:

1. **Random selection performs well**: Our default strategy of random selection achieves strong performance on both tasks, validating the practical applicability of our method without requiring careful sample curation.

2. **Correctly predicted samples can enhance performance**: On BoolQ, using samples that the model already predicts correctly leads to a 2.9% improvement over random selection, suggesting that DSM neurons identified from high-confidence correct predictions effectively capture spurious patterns.

3. **Incorrectly predicted samples underperform**: Using only misclassified samples results in lower performance (79.81% on BoolQ, 87.5% on SST2), indicating that errors may stem from various sources beyond DSM neurons, making them less effective for identifying systematic shortcuts.

4. **Task-dependent effects**: The impact of sample selection varies by task. SST2 shows identical performance for random and correct samples, while BoolQ shows more sensitivity, suggesting that the optimal selection strategy may be task-specific.

These findings demonstrate that our DSM neuron pruning method exhibits low sensitivity to the random selection of adaptation samples, making it practical for real-world applications where careful sample curation may not be feasible. The low variance ($\pm 0.65\%$) across different random seeds and the strong performance with random selection validate the robustness of our approach.

### E.2 Justification for Single-Layer Pruning Strategy

Our approach of pruning within a single layer (specifically the gate_proj layer) with a grid search over pruning percentages is grounded in both empirical evidence and practical considerations. This section provides detailed justification for these design choices.

**Empirical Validation of Layer Selection** To validate our focus on the gate_proj layer, we conducted comparative experiments across all MLP components in LLAMA3.1-8B-Instruct. Our DSM neuron detection and pruning method was applied independently to each MLP layer type (gate_proj, up_proj, and down_proj) to evaluate the resulting performance.

| Dataset | Baseline | gate_proj | up_proj | down_proj |
|---------|----------|-----------|---------|-----------|
| BoolQ | 75.19% | **82.35%** | 81.84% | 79.53% |
| SST2 | 88.87% | **92.14%** | 91.02% | 90.36% |

Table 9: Performance comparison of pruning different MLP layers in LLAMA3.1-8B-Instruct. Bold indicates best performance.

The results demonstrate that pruning the gate_proj layer consistently yields the highest performance gains. On BoolQ, gate_proj achieves +7.16% improvement, compared to +6.65% for up_proj and +4.34% for down_proj. Similarly, on SST2, gate_proj achieves +3.27% improvement, compared to +2.15% for up_proj and +1.49% for down_proj.

This superior performance of gate_proj pruning aligns with recent findings in the interpretability literature. The gating mechanism plays a crucial role in selecting which knowledge-related behaviors to activate (Ali et al., 2024), making it a natural location where domain-specific mechanisms might be encoded. DSM neurons that rely on spurious correlations may preferentially manifest in this gating layer, as it controls the flow of information through the MLP block.

### E.3 Sensitivity to Pruning Threshold

To assess the robustness of our method to the pruning threshold selection, we conducted a detailed analysis of performance across different pruning percentages. Using LLAMA3.1-8B-Instruct, we evaluated the impact of varying the pruning percentage from 0% (baseline) to 40% on both BoolQ and SST2 datasets.

| Dataset | 0% | 10% | 15% | 20% | 25% | 30% | 35% | 40% |
|---------|-----|------|------|------|------|------|------|------|
| BoolQ | 75.19 | 80.80 | 81.46 | 81.28 | 80.90 | 81.00 | **82.32** | 79.12 |
| SST2 | 88.87 | 89.92 | 90.12 | 91.45 | **92.14** | 92.05 | 92.10 | 90.12 |

Table 10: Performance of LLAMA3.1-8B-Instruct across different pruning percentages on BoolQ and SST2. Bold indicates peak performance for each dataset.

The results demonstrate that our method exhibits robust performance across a wide range of pruning thresholds. For BoolQ, all pruning percentages from 10% to 35% yield substantial improvements over the baseline (75.19%), with performance ranging from 80.80% to 82.32%. The peak performance of 82.32% is achieved at 35% pruning, representing a 7.13% absolute improvement. Notably, the performance remains consistently high (above 80%) across most pruning percentages, only dropping below this threshold at the extreme of 40% pruning.

Similarly, SST2 shows consistent improvements across all tested pruning percentages. Starting from a baseline of 88.87%, performance improves to between 89.92% and 92.14%, with the peak at 25% pruning. The performance remains above 90% for pruning percentages between 15% and 35%, demonstrating a stable improvement window.

This analysis confirms that our pruning approach is robust to threshold selection, making it practical for deployment without extensive hyperparameter tuning.

