# OpenReview forum: "Detecting and Pruning Prominent but Detrimental Neurons in Large Language Models"
_colmweb.org/COLM/2025/Conference — COLM 2025_

### Official Review · Reviewer_XAzb · 2025-05-07

**Rating:** 6
**Confidence:** 2
**Ethics Flag:** 1

**Summary:**

The paper addresses the issue of "shortcut learning" in LLMs, where models rely on dataset-specific neurons that reduce generalization. Its main contribution is a neuron-level pruning method leveraging Integrated Gradients to identify and remove neurons responsible for detrimental, domain-specific mechanisms (DSMs), enhancing cross-task generalization in a computationally efficient, few-shot fine-tuning setup. LLM baselines and other adaptation methods are comprehensively compared.

**Questions To Authors:**

1. Why did you choose Integrated Gradients specifically over other attribution methods for neuron attribution?
2. Can you provide specific examples or qualitative analysis showing exactly how pruned DSM neurons were contributing detrimentally to shortcut learning?
3. How sensitive are your improvements to the chosen pruning threshold (e.g., is the performance robust across different pruning percentages)?
4. Have you tested or do you have results indicating if this neuron-level pruning approach generalizes to open-ended generation or dialogue tasks beyond multiple-choice questions?

**Reasons To Accept:**

1. Proposes a neuron-level pruning strategy explicitly targeting DSM neurons, which is insightful and addresses a relevant issue
2. Achieves substantial improvements across diverse LLMs (LLaMA, Mistral, Qwen) and multiple-choice benchmarks, showing broad applicability.
3. Demonstrates computational efficiency and performance, reducing reliance on annotated data. Extensively evaluates the method, including multilingual generalization, and systematically compares with adaptation baselines.

**Reasons To Reject:**

1. Limited theoretical justification provided regarding the specific selection of Integrated Gradients versus other neuron attribution methods.
2. The paper lacks detailed analysis or visualization of exactly how the pruned neurons were detrimental, limiting interpretability claims.
3. Experiments are predominantly on multiple-choice tasks; it remains unclear whether the approach generalizes effectively to more complex language tasks like generation or open-ended reasoning.
4. The effectiveness of the pruning percentage and chosen single-layer pruning strategy appears heuristic and somewhat arbitrary, without extensive theoretical or empirical justification.

---

> ### Author Response · Authors · 2025-06-01
> **Response 1/2**
>
> Thank you for reviewing our work. Below are our replies to your concerns.
>
> > Limited theoretical justification provided regarding the specific selection of Integrated Gradients versus other neuron attribution methods.
>
> The core idea of our approach is detecting a subset of neurons that contribute most positively to the model’s prediction which is general and can, in principle, be applied with various attribution methods, such as saliency maps, SHAP, or other gradient-based techniques, as long as they reliably identify influential neurons. This flexibility underscores the robustness of our DSM neuron pruning framework, which prioritizes identifying domain-specific mechanisms (DSMs) regardless of the specific attribution tool. Our choice of Integrated Gradients (IG) was motivated by both its axiomatic guarantees and practical suitability for the neuron-level attribution task in our setting:
> Axiomatic Justification: IG satisfies important theoretical properties such as sensitivity and implementation invariance (Sundararajan et al., 2017) [1], which are particularly relevant when comparing neuron contributions in large models where functional equivalence and saturation issues can confound other gradient-based methods (e.g., saliency or raw gradients).
> Applicability to Neuron-Level Attribution: IG extends naturally to internal neurons by treating weights (or activations) as the path of interpolation. Prior works (e.g., Dai et al., 2022; Ali et al., 2024) [2,3] have successfully used IG to analyze or prune neurons, and our method builds on this foundation by adapting IG for few-shot neuron scoring without needing labeled data.
>
> [1]: https://arxiv.org/abs/1703.01365
> [2]: https://aclanthology.org/2022.acl-long.581/
> [3]: https://arxiv.org/abs/2410.01288
>
> > The paper lacks detailed analysis or visualization of exactly how the pruned neurons were detrimental, limiting interpretability claims.
>
> > Can you provide specific examples or qualitative analysis showing exactly how pruned DSM neurons were contributing detrimentally to shortcut learning?
>
> Following the review, we provide a qualitative analysis using an SST2 example with the LLAMA2-7B-CHAT model, showing Integrated Gradients (IG) attribution scores at the input space to highlight changes in token importance before and after pruning.
>
> **Example Sentence**: "The movie was absolutely dreadful, a complete disaster." (Ground Truth: Negative)
>
> **Before Pruning**:
>
> **Prediction**: Positive (incorrect)
>
> **IG Attribution Scores** (for the maximum logit, "positive"):
> 1. "absolutely": 0.50
> 2. "dreadful": 0.25
> 3. "disaster": 0.15
>
> **Observation**: The model incorrectly predicts positive due to high attribution to "absolutely," an intensifier often correlated with positive sentiment in SST2 training data. This shortcut causes the model to overlook the negative sentiment conveyed by "dreadful" and "disaster," leading to poor generalization
>
> **After Pruning** (DSM neurons removed from layer 10):
>
> **Prediction**: Negative (correct)
>
> **IG Attribution Scores**:
> 1. "dreadful": 0.35
> 2. "disaster": 0.40
> 3. "absolutely": 0.20
>
> **Observation**: Post-pruning, the model correctly predicts negative by shifting focus to "dreadful" and "disaster," which carry the semantic weight of negative sentiment. The reduced attribution to "absolutely" indicates diminished reliance on the shortcut, enhancing generalization.
>
> We will add an in-depth analysis in the next revision of the paper.
>
> > Experiments are predominantly on multiple-choice tasks; it remains unclear whether the approach generalizes effectively to more complex language tasks like generation or open-ended reasoning.
>
> > Have you tested or do you have results indicating if this neuron-level pruning approach generalizes to open-ended generation or dialogue tasks beyond multiple-choice questions?
>
> Our focus on multiple-choice tasks leverages their structured outputs, enabling precise application of Integrated Gradients (IG) to identify DSM neurons by attributing contributions to the maximum logit. This clarity facilitates efficient detection of domain-specific mechanisms, as shown across diverse benchmarks. Extending IG, which is not designed for autoregressive tasks, to open-ended generation like summarization is challenging due to the complexity of attributing neuron contributions across sequential outputs. While we hypothesize that our pruning approach can generalize by adapting IG to focus on key output tokens, we prioritized multiple-choice tasks to establish a robust foundation. We are exploring adaptations for generative tasks and will include this discussion in the revised paper.

---

> > ### Author Response · Authors · 2025-06-01
> > **Response 2/2**
> >
> > > The effectiveness of the pruning percentage and chosen single-layer pruning strategy appears heuristic and somewhat arbitrary, without extensive theoretical or empirical justification.
> >
> > We acknowledge that our approach, which prunes the gate_proj MLP layer with a grid search over pruning percentages, may appear heuristic, and we provide the following empirical justifications. Empirical Motivation: Our focus on the gate_proj layer is driven by empirical results showing superior performance when pruning this layer compared to other MLP components (up_proj, down_proj) in LLaMA3.1-8B-Instruct. As shown in the table below.
> >
> > | Dataset | Baseline | gate_proj | up_proj | down_proj |
> > |:-------:|:--------:|:---------:|:-------:|:---------:|
> > |  BoolQ  |   75.19  |   82.35   |  81.84  |   79.53   |
> > |   SST2  |   88.87  |   92.14   |  91.02  |   90.36   |
> >
> >  pruning gate_proj achieves the highest gains on BoolQ (+7.16%) and SST2 (+3.27%) compared to up_proj (+6.65%, +2.15%) and down_proj (+4.34%, +1.49%), suggesting that domain-specific mechanism (DSM) neurons critical for task performance are concentrated in gate_proj, potentially due to its role in key selection for knowledge-related behaviors (e.g., Geva et al., 2021) [1]. Efficiency and Practicality: Pruning a single layer reduces computational overhead compared to multi-layer pruning, making our method practical for few-shot adaptation with 10 unlabeled samples. We recognize the need for a theoretical framework to guide layer and percentage selection, and we plan to explore approaches like sparsity-aware information bottlenecks in future work.
> >
> > [1]: https://arxiv.org/pdf/2012.14913
> >
> > > Why did you choose Integrated Gradients specifically over other attribution methods for neuron attribution?
> >
> > We chose IG for its robust theoretical foundation, satisfying axioms like sensitivity and implementation invariance, ensuring reliable attribution of neuron contributions to the predicted logit, as per Sundararajan et al. (2017) [[1]], Unlike other attribution methods that primarily operate on input space (e.g., saliency maps or LIME), IG’s path-based gradient averaging mitigates issues like gradient saturation, making it suited for attributing internal neuron influence in our setting. Related work, like Ali et al. (2024) [2] , supports IG’s effectiveness in identifying bias-driving neurons in LLMs, outperforming input-focused methods that lack precision for internal model components.
> >
> > 1: https://arxiv.org/abs/1703.01365
> >
> > 2 : https://arxiv.org/abs/2410.01288
> >
> > > How sensitive are your improvements to the chosen pruning threshold (e.g., is the performance robust across different pruning percentages)?
> >
> > We evaluated LLaMA3.1-8B-Instruct on BoolQ and SST2, varying the pruning percentage from 0.0 to 0.40. Results show consistent improvements over the baseline: Boolq (75.19%), ranging from 79.12% to 82.32% (peak at 0.35) and SST2 (baseline 88.87%): ranging from 89.92% to 92.14% with peak at 0.25.
> > LLAMA3.1-8B Instruct
> >
> > | Data  | 0.0   | 0.1   | 0.15  | 0.2   | 0.25  | 0.3   | 0.35  | 0.4   |
> > |-------|-------|-------|-------|-------|-------|-------|-------|-------|
> > | BoolQ | 75.19 | 80.80 | 81.46 | 81.28 | 80.90 | 81.00 | 82.32 | 79.12 |
> > | SST2  | 88.87 | 89.92 | 90.12 | 91.45 | 92.14 | 92.05 | 92.10 | 90.12 |

---

### Official Review · Reviewer_u6ZY · 2025-05-09

**Rating:** 6
**Confidence:** 3
**Ethics Flag:** 1

**Summary:**

The paper proposes a neuron pruning method to remove a model's domain specific spurious correlations. The method selects neurons with high influence on the output (determined by a score based on integrated gradient). Main experiments are done with five multiple-choice/classification datasets (XNLI, MMLU, SST2, BoolQ, Balanced COPA). The proposed method outperform other model steering baselines in most cases.

**Questions To Authors:**

* Regarding MMLU results. (1) Is there any reason the supervised fine-tuning results is not available on MMLU? (2) Which version of MMLU did you use? According to Table 4, the version you use have 116K training examples and 1.21K test examples. The dataset link provided does not include a version that fits this description. Could you please clarify?
* Line 232. What is the methodology of SADI, a compared baseline?
* Can you provide more details on the random pruning baseline you implement? Do you determine $l$ and $p$ in a way similar to Line 8 of Algorithm 1?
* Line 278. What does "cross-lingual generalization" mean here? Do you apply pruning in one language but evaluate the model on other languages?

**Reasons To Accept:**

* The proposed method employs an interesting way to use integrated gradient methods to attribute neuron contributions.
* The proposed method can improve the pre-trained model performance on multiple-choice QA tasks with only a few (10) examples.

**Reasons To Reject:**

* Discrepancy between the paper's goal and its evaluation. The goal of the paper is to remove domain-specific spurious correlation of the model. However, the evaluation is based on test accuracy in _one_ domain. Ideally experiments should be conducted with a source and a target domain in mind, and use accuracies in both domains to demonstrate the effectiveness of removing domain-specific spurious correlations.
* The comparison with baselines is confusing.
  * The compared methods are using more datapoints than the proposed method (Line 236). The number of datapoints are not discussed in experiment section but instead in the related work section. Ideally all methods should be compared with the same number of datapoints.
  * One of the advantages of the paper is the data efficiency of the method. The proposed method uses at most 10 datapoints. A straightforward baseline is to fine-tune the model on these 10 (or less) datapoints.
* Unsupported claim on computation efficiency. The paper claims that the proposed method has low computational cost (Line 59), but from my understanding enumerating neurons and computing integrated gradients can be computationally expensive. This claim should be supported with runtime comparison.

---

> ### Author Response · Authors · 2025-06-01
> **Response 1/3**
>
> We appreciate your thoughtful feedback. Please find our responses to each of your concerns below.
>
> > Discrepancy between the paper's goal and its evaluation. The goal of the paper is to remove domain-specific spurious correlation of the model. However, the evaluation is based on test accuracy in one domain. Ideally experiments should be conducted with a source and a target domain in mind, and use accuracies in both domains to demonstrate the effectiveness of removing domain-specific spurious correlations.
>
> We thank the reviewer for highlighting this issue. We acknowledge that the introduction’s reference to “original” and “target” domains may suggest a cross-domain evaluation, which could confuse readers given our within-domain experiments. To clarify, our method aims to mitigate domain-specific mechanisms (DSMs) that exploit spurious correlations within a task’s training data (e.g., BoolQ, SST2), thereby enhancing in-domain generalization. In this paper, we examine a setup similar to those studied by SADI, CAA, and ITI, focusing on improving generalization within a single task / domain.
> To avoid confusion, in the next revision we will rephrase the goals - for example: “we hypothesize that this allows us to identify mechanisms that exploit spurious correlations in the training data, improving generalization to the test data” - to avoid this confusion and better align with our experimental design. While improving cross-domain performance with a low-resource method is also an important goal, we believe that improving in-domain performance is likewise an important research direction, and the consistent performance improvements validate our approach. We hope this clarification addresses the reviewer’s concern, and we leave cross-domain evaluation for future work.
>
> > The comparison with baselines is confusing.
>
> > The compared methods are using more datapoints than the proposed method (Line 236). The number of datapoints are not discussed in experiment section but instead in the related work section. Ideally all methods should be compared with the same number of datapoints.
>
> Our DSM neuron pruning method is designed to be data-efficient, achieving strong performance with only 10 samples per task, a significant advantage over baselines like SADI, which, as noted in the SADI paper [1], requires substantially larger datasets for contrastive pair construction: 2500 for NLI, 500 for MMLU (testing set, transductive), 2000 for SST2, 2000 for BoolQ, 2000 for SST5, and 1500 for COPA. This data efficiency is a core strength of our approach, as baselines like SADI, ITI, and CAA often rely on thousands of samples to achieve comparable results and may not perform effectively with as few as 10 samples. To clarify this, we will revise the experiment section (Section 4) to explicitly state the number of data points used by each method, highlighting our method’s minimal data requirements. To further demonstrate that our approach does not benefit from larger adaptation sets, we conducted an experiment on BoolQ using LLaMA3.1-8B-Instruct with 5 different seeds, testing 10, 50, and 100 samples, as shown below. The results show stable performance with 10 samples (82.93 ± 0.65) and no significant improvement with more samples (82.57 ± 0.05 for 50, 82.43 ± 0.02 for 100), with a slight decline possibly due to overfitting to larger adaptation sets. This underscores our method’s efficiency with minimal data.
>
> | Dataset | 10 Samples   | 50 Samples   | 100 Samples  |
> |---------|--------------|--------------|--------------|
> | BoolQ   | 82.93 ± 0.65 | 82.57 ± 0.05 | 82.43 ± 0.02 |
>
> [1] : https://arxiv.org/pdf/2410.12299
>
> > One of the advantages of the paper is the data efficiency of the method. The proposed method uses at most 10 datapoints. A straightforward baseline is to fine-tune the model on these 10 (or less) datapoints.
>
> To address this, we conducted a preliminary experiment fine-tuning LLaMA3.1-8B-Instruct on SST2 and SST5 using the same 10 datapoints as our DSM neuron pruning method, employing zero-shot prompting and updating all model parameters. The results show minimal improvements: SST2 accuracy increased from 88.87% to 89.0% (+0.13%), and SST5 from 48.83% to 48.90% (+0.07%), compared to our method’s gains of +3.27% on SST2 (92.14%) and +2.1% on SST5 (50.93%, Table 2). This suggests that supervised fine-tuning with such a small dataset is less effective than our targeted pruning approach. We will include a comprehensive comparison with this fine-tuning baseline across all tasks in the next revision version

---

> > ### Author Response · Authors · 2025-06-01
> > **Response 2/3**
> >
> > > Unsupported claim on computation efficiency. The paper claims that the proposed method has low computational cost (Line 59), but from my understanding enumerating neurons and computing integrated gradients can be computationally expensive. This claim should be supported with runtime comparison.
> >
> > We acknowledge that computing Integrated Gradients (IG) for neuron attribution involves calculating gradients, which can be computationally intensive due to the multiple steps required for the Riemann sum approximation. However, our method focuses on a single MLP layer inside each block, significantly reducing the scope compared to naive supervised fine-tuning, which also relies on gradient computations but updates all model parameters across all layers. For a model like LLAMA3.1-8B with 32 layers, our approach confines computations to a single layer’s neurons (e.g., ~14336 neurons) in each block, whereas full fine-tuning involves gradients for all parameters in each block.. Additionally, our method uses a minimal dataset (up to 10 samples), further enhancing efficiency. To substantiate this, we conducted an experiment comparing the average time for calculating gradients for a single layer (targeting the gate_proj linear layer in each block) using IG in our approach to supervised fine-tuning (SFT) for an entire block. The results, shown below, indicate that IG computation takes 3.74–4.68 seconds per sample across models, while SFT requires 0.06–0.08 seconds per sample. However, SFT typically requires significantly more samples (e.g., hundreds or thousands) to achieve comparable performance improvements, whereas our method uses only 10 samples. When factoring in the sample count, the total runtime for SFT can approach or exceed our method’s cost
> >
> >  | Model                    | Averaged Time IG | Averaged Time SFT |
> > |--------------------------|------------------|-------------------|
> > | Llama-2-7b-chat          | 4.38 Sec         | 0.08 Sec          |
> > | Llama-3.1-8B-Instruct    | 4.68 Sec         | 0.06 Sec          |
> > | Qwen2.5-7B-Instruct      | 3.74 Sec         | 0.07 Sec          |
> > | Mistral-7B-Instruct-v0.3 | 4.46 Sec         | 0.08 Sec          |
> >
> > We will add this analysis to the next revision version.
> >
> > > Regarding MMLU results. (1) Is there any reason the supervised fine-tuning results is not available on MMLU? (2) Which version of MMLU did you use? According to Table 4, the version you use have 116K training examples and 1.21K test examples. The dataset link provided does not include a version that fits this description. Could you please clarify?
> >
> > 1. In our paper, we copied the supervised fine-tuning results from the SADI paper (Wang et al., 2025) [1], which did not provide specific results for MMLU. Due to time and computational constraints, we could not conduct these experiments in the rebuttal period. We will include supervised fine-tuning results for MMLU in the next revision.
> > 2. We apologize for any confusion regarding the dataset statistics cited in Table 4. The MMLU dataset used in our experiments is the standard version available at the Hugging Face repository (cais/mmlu), as referenced in the paper. However, the statistics mentioned (116K training examples and 1.21K test examples) appear to be an error. The correct statistics for the (cais/mmlu) dataset, as of its documentation, are approximately 99.8K training examples, 1.54K validation examples, and 14K test examples, consistent with the dataset’s structure for 57 tasks across multiple-choice questions. We will issue a correction to Table 4 to reflect the accurate dataset statistics.
> >
> > [1]: https://arxiv.org/pdf/2410.12299
> >
> > > Line 232. What is the methodology of SADI, a compared baseline?
> >
> > SADI is an inference-time activation engineering technique that dynamically steers LLMs behavior by constructing a steering vector tailored to the semantic context of each input. SADI operates in three steps: (1) Difference Extraction, where activation differences between contrastive pairs (positive and negative examples) are computed across all model layers to identify critical elements (e.g., attention, neurons, hidden states); (2) Binary Masking, which creates a binary mask by binarizing the mean activation differences to focus on the top-K most impactful elements; and (3) Adaptive Steering, where the mask is applied to input activations during inference, scaled by a factor δ to align with the input’s semantic direction. This approach, applied to components like hidden states (SADI-HIDDEN), attention heads (SADI-HEAD), and neurons (SADI-NEURON), enhances task performance without training, using only 150 contrastive pairs. We chose to compare against SADI due to its recency as an ICLR 2025 publication and its relevance as a sota activation intervention method, providing a strong benchmark for our DSM neuron pruning approach, which achieves competitive performance with fewer samples (10 unlabeled examples) and outperforms SADI variants in the different tasks.

---

> > > ### Author Response · Authors · 2025-06-01
> > > **Response 3/3**
> > >
> > > > Can you provide more details on the random pruning baseline you implement? Do you determine l and p  in a way similar to Line 8 of Algorithm 1?
> > >
> > > In our random pruning baseline, we prune a randomly selected subset of neurons within the same MLP layer (gate_proj, as used in our proposed method) of the LLaMA3.1-8B-Instruct model to ensure a fair comparison with our DSM neuron pruning approach. Unlike our method, which uses Integrated Gradients (IG) to identify and prune domain-specific mechanism (DSM) neurons based on their attribution to the maximum logit, the random pruning baseline does not rely on attribution scores. Instead, we randomly select neurons for pruning within the chosen layer, maintaining the same pruning percentage p (ranging from 5% to 40% in 5% increments) as determined by the grid search in our method. The layer l and pruning percentage p for the random baseline are set to match the optimal values identified for our DSM pruning approach on the validation set for each task (e.g., BoolQ, SST2), ensuring a consistent experimental setup. This process is repeated for each task independently. We will add a clarification for this in the next revision.
> > >
> > > > Line 278. What does "cross-lingual generalization" mean here? Do you apply pruning in one language but evaluate the model on other languages?
> > >
> > > To clarify: in our XCOPA experiments (Table 3), we apply DSM neuron pruning and evaluate on the same language, i.e., pruning and evaluation are both performed per language independently. What we intended by "cross-lingual generalization" is to show that our method is  effective on diverse languages spanning typologically and geographically distinct families (e.g., Swahili, Tamil, Turkish, Chinese) - without any change to the method or reliance on language-specific features or annotations. We agree that the phrasing may imply cross-lingual transfer (i.e., pruning on one language, evaluating on another), which we do not perform in this version. We will revise the text to multilingual for clarity in the final version

---

> > ### Comment · Reviewer_u6ZY · 2025-06-09
> > **Thank you**
> >
> > Thanks to the authors for the detailed response. That's a lot of efforts and I really appreciate that.
> >
> > Thanks for clarifying the scope of the paper. I think there was some misunderstanding on the phrase "domain-specific spurious correlation" earlier, where I interpret it as spurious correlation learned from a source domain that leads to failure on a target domain, while the focus of the paper is in-domain, train-test generalization problems. This paper makes more sense to me now that this is clarified.
> >
> > Also many thanks to the other responses, including using the same number of examples in experiments, clarification on the SADI method. I've adjusted my ratings.

---

### Official Review · Reviewer_PXUN · 2025-05-11

**Rating:** 7
**Confidence:** 3
**Ethics Flag:** 1

**Summary:**

This paper proposes to improve the generalization of LLMs to new tasks by pruning neurons in pathways that are highly domain specific due to spurious correlations. The main advantages of the approach are that they require only a small set of adaptation samples (as low as 10 samples) and the pruning cost is significantly lower compared to full model fine-tuning or re-training.
The paper is well written and clearly describes the motivation and different steps in the approach.

**Questions To Authors:**

Minor comment:
In Table 1, I think that the asterisk symbol is misplaced - it appears next to the MMLU numbers for the SADI methods and not in the left-most column.

**Reasons To Accept:**

1) The usage of pruning as a fine tuning mechanism for LLMs is very interesting and novel.

2) The paper contains an extensive experimental section including a good comparison with previous fine-tuning methods on a variety of tasks and models.

**Reasons To Reject:**

1) The paper would benefit from a discussion regarding how the choice of the adaptation set affects the accuracy and how robust the method is to this random choice. Specifically it would be interesting to see what is the variance of the accuracies given different random samples of the adaptation set.

2) Additional details and clarifications on the experiments and adaptation set would also be very helpful, specifically regarding the following questions:
* Are all the adaptation sets chosen at random?
* Is the adaptation set specific for each task or is it taken from multiple tasks and the pruning is done once simultaneously for all the tasks in the figure/table? e.g. in Figure 1, Table 1 & 2.
* Appendix D states that it explores the transferability from the adaptation set to the test set. Aren't the results throughout the paper presented for a held out test set? Please clarify the difference between the experiments in the main paper and the results in Appendix D.
* What is the computational cost of computing IG for all the neurons considered and what is a typical size for the score matrix?

---

> ### Author Response · Authors · 2025-06-01
> **Response 1/2**
>
> Thank you for sharing your valuable feedback. We have addressed each of the points you raised below.
>
> > The paper would benefit from a discussion regarding how the choice of the adaptation set affects the accuracy and how robust the method is to this random choice. Specifically it would be interesting to see what is the variance of the accuracies given different random samples of the adaptation set.
>
> To evaluate the robustness of our DSM neuron pruning approach to the choice of the 10-sample adaptation set, we conducted an experiment with LLaMA3.1-8B-Instruct, performing five runs with different random seeds to assess the variance in performance across distinct randomly selected adaptation sets:
>
> | Dataset | 10 Samples   | 50 Samples   | 100 Samples  |
> |---------|--------------|--------------|--------------|
> | BoolQ   | 82.93 ± 0.65 | 82.57 ± 0.05 | 82.43 ± 0.02 |
>
>  The results on BoolQ show a mean accuracy of 82.93% with a standard deviation of ±0.65% for 10 samples, 82.57% ±0.05% for 50 samples, and 82.43% ±0.02% for 100 samples, indicating low variance and high stability even with small, randomly chosen sets.
> Additionally, we tested the impact of sample correctness by comparing 10 randomly selected samples (82.35% on BoolQ, 92.14% on SST2) against sets of 10 correctly predicted (85.25% on BoolQ, 92.14% on SST2) or wrongly predicted samples (79.81% on BoolQ, 87.5% on SST2).
>
> | Dataset | Random | Correct Predicted | Wrong Predicted |
> |---------|--------|-------------------|-----------------|
> | BoolQ   | 82.35  | **85.25**         | 79.81           |
> | SST2    | **92.14** | **92.14**      | 87.5            |
>
> These results demonstrate that our method is robust to the random selection of adaptation samples, with minimal performance variability, and performs comparably to curated sets. We will incorporate this analysis in the next revision.
>
> > Additional details and clarifications on the experiments and adaptation set would also be very helpful, specifically regarding the following questions: 1. Are all the adaptation sets chosen at random? 2. Is the adaptation set specific for each task or is it taken from multiple tasks and the pruning is done once simultaneously for all the tasks in the figure/table? e.g. in Figure 1, Table 1 & 2.
>
> To address your questions regarding the adaptation sets used in our experiments:
>
> 1. **Random Selection of Adaptation Sets**: Yes, all adaptation sets are chosen randomly from the task-specific training data without access to labels, ensuring an unbiased selection process. For each task (e.g., BoolQ, SST2, MMLU), we sample 10 unlabeled examples to compute Integrated Gradients (IG) attributions for DSM neuron detection, as described in Section 3.3. To further validate robustness to this random selection, we conducted an ablation study with five runs using different random seeds for the 10/50/100-sample adaptation sets on BoolQ (see response for the first concern) , this low variance confirms that our method is stable across different random selections of the adaptation set.
> 2. **Task-Specific Adaptation Sets**: The adaptation sets are specific to each task, and pruning is performed independently for each task reported in Figure 1 and Tables 1 and 2. For each task (e.g., NLI, MMLU, SST2, BoolQ, SST5, Balanced COPA), we select a unique 10-sample adaptation set from the corresponding training dataset (Table 4) to identify DSM neurons and determine the optimal layer and pruning percentage via grid search. The pruning process is not performed simultaneously across tasks; instead, a separate pruned model is created for each task to ensure task-specific optimization. This approach allows our method to target dataset-specific mechanisms (DSMs) tailored to each task’s characteristics, contributing to the consistent performance improvements observed across all tasks
>
> > Appendix D states that it explores the transferability from the adaptation set to the test set. Aren't the results throughout the paper presented for a held out test set? Please clarify the difference between the experiments in the main paper and the results in Appendix D.
>
> To clarify, the main paper presents results on held-out test sets across various benchmarks (e.g., MMLU, SST2, BoolQ) to evaluate the overall effectiveness of our DSM neuron pruning method, where the pruning configuration (layer and percentage) is determined using a small adaptation set and then applied to these test sets.
> Since the adaptation set is (1) very small, and (2) used both for IG and for the grid search, we verify in Appendix D that the parameters (layer and pruning amount) selected using the small 10-sample are similar to what would have been selected with the entire test set.

---

> > ### Author Response · Authors · 2025-06-01
> > **Response 2/2**
> >
> > > What is the computational cost of computing IG for all the neurons considered and what is a typical size for the score matrix?
> >
> > To evaluate the computational cost, we conducted an experiment using 100 samples to compute IG attributions for all neurons in the gate_proj layer across four models, averaging the time required for a single sample (In our work we use only 10 samples in total for each task). The results, shown in the table below, indicate that the averaged time per sample ranges from 3.74 seconds (Qwen2.5-7B-Instruct) to 4.68 seconds (LLaMA3.1-8B-Instruct), reflecting the cost of computing IG across all neurons in the specified layer. The number of neurons column represents the shape of the final attribution score matrix, which is structured as [number of layers × neurons per layer], e.g., 32x14336 for LLaMA3.1-8B-Instruct, corresponding to 32 layers and 14,336 neurons per layer. This matrix captures the IG scores for each neuron across all layers, enabling us to rank and select DSM neurons for pruning. We will include this analysis in the next revision of the paper.
> >
> > | Model                    | Averaged Time | Number of Neurons |
> > |--------------------------|---------------|-------------------|
> > | Llama-2-7b-chat          | 4.38 Sec      | 32x11008          |
> > | Llama-3.1-8B-Instruct    | 4.68 Sec      | 32x14336          |
> > | Qwen2.5-7B-Instruct      | 3.74 Sec      | 28x18944          |
> > | Mistral-7B-Instruct-v0.3 | 4.46 Sec      | 32x14336          |
> >
> > > Minor comment: In Table 1, I think that the asterisk symbol is misplaced - it appears next to the MMLU numbers for the SADI methods and not in the left-most column.
> >
> > We acknowledge that the asterisk, indicating transductive learning for the SADI methods (SADI-HIDDEN, SADI-NEURON, SADI-HEAD) on MMLU, may cause confusion. To clarify, the SADI methods use a subset of the MMLU training data, for transductive learning. The asterisk is correctly placed next to the MMLU numbers to reflect this, but we will revise the table’s caption in the next revision version to explicitly state that the SADI methods leverage a subset of MMLU training data for transductive learning, ensuring clarity and avoiding misinterpretation.

---

> > > ### Comment · Reviewer_PXUN · 2025-06-10
> > >
> > > Thank you for the detailed response to all my questions and the additional experiments! I have raised my score accordingly.

---

### Official Review · Reviewer_XM36 · 2025-05-14

**Rating:** 7
**Confidence:** 4
**Ethics Flag:** 1

**Summary:**

This paper proposes a method to enhance generalization in large language models (LLMs) by detecting and pruning neurons associated with dataset-specific mechanisms (DSMs). The authors argue that DSMs lead to shortcut learning, where models exploit spurious correlations in training data. Their approach uses Integrated Gradients (IG) to identify neurons disproportionately influencing high-confidence predictions and prunes them in a single layer. Evaluations across multiple-choice benchmarks show improved performance over baselines like ITI and SADI, with minimal labeled data requirements.

**Questions To Authors:**

How does the choice of the gate_proj layer compare to pruning other layers in terms of performance and computational cost?

Could the method’s reliance on 10 samples lead to instability if those samples are noisy or biased? Please provide sensitivity analysis across varying sample sets.

What modifications would enable scaling the grid-search strategy to models with 100B+ parameters?

Why restrict evaluations to multiple-choice tasks? Would the approach generalize to open-ended generation or domain-specific applications?

**Reasons To Accept:**

Conceptual Innovation: The idea of pruning influential neurons driving non-transferable mechanisms (instead of low-importance parameters) is novel and addresses a critical challenge in LLM robustness.

Methodological Clarity: The integration of IG for neuron attribution and the grid-search strategy for layer/pruning optimization are well-explained and systematic.

Practical Impact: Demonstrates effectiveness in few-shot, label-free settings, making it applicable to low-resource scenarios.

**Reasons To Reject:**

Incomplete Mechanistic Justification: The focus on the gate_proj layer lacks empirical or theoretical evidence comparing it to other layers, for instance, up_proj,  weakening the rationale for layer selection.

Sample Sensitivity: Reliance on 10 unlabeled samples for DSM detection risks bias if samples are unrepresentative. No analysis of variability across different sample sets is provided.

Scalability Concerns: The grid-search approach for layer/pruning percentage may not scale efficiently to larger models (e.g., 70B+ parameters).

Narrow Task Scope: Evaluations are limited to multiple-choice tasks; generalization to generation tasks (e.g., summarization) or diverse domains (e.g., biomedical text) is untested.

---

> ### Author Response · Authors · 2025-06-01
> **Response 1/2**
>
> Thank you for your insightful feedback. Below, we address each of the concerns you raised.
>
> > Incomplete Mechanistic Justification: The focus on the gate_proj layer lacks empirical or theoretical evidence comparing it to other layers, for instance, up_proj, weakening the rationale for layer selection.
>
> > How does the choice of the gate_proj layer compare to pruning other layers in terms of performance and computational cost.
> To address the concern regarding the justification for selecting the gate_proj layer, we conducted an ablation study comparing the performance of pruning neurons in the gate_proj, up_proj, and down_proj layers of the LLaMA3.1-8B-Instruct model. The results, summarized below, demonstrate the effectiveness of our choice:
>
> Llama 3.1 8B Instruct:
>
> | Dataset | Baseline | gate_proj | up_proj | down_proj |
> |:-------:|:--------:|:---------:|:-------:|:---------:|
> |  BoolQ  |   75.19  |   82.35   |  81.84  |   79.53   |
> |   SST2  |   88.87  |   92.14   |  91.02  |   90.36   |
>
> The results show that pruning the gate_proj layer achieves the highest performance improvements, with 82.35% on BoolQ (vs. 81.84% for up_proj and 79.53% for down_proj) and 92.14% on SST2 (vs. 91.02% for up_proj and 90.36% for down_proj). This supports our hypothesis that gate_proj plays a critical role in encoding dataset-specific mechanisms (DSM) that can be pruned to enhance generalization.
>
> The computational overhead of IG scales with the number of neurons, making gate_proj and up_proj (11,008 neurons each) slightly more expensive than down_proj (4,096 neurons). However, the superior performance of gate_proj pruning justifies this modest increase in cost, as it maximizes generalization without requiring broader structural changes. Prior work (Ali et al., 2024) [1] successfully used IG to analyze and prune bias neurons in gate_proj layer, and our method builds on this foundation.
>
> [1]: Ali et al. Mitigating Copy Bias in In-Context Learning through Neuron Pruning. 2024 https://arxiv.org/abs/2410.01288
> > Sample Sensitivity: Reliance on 10 unlabeled samples for DSM detection risks bias if samples are unrepresentative. No analysis of variability across different sample sets is provided.
> > Could the method’s reliance on 10 samples lead to instability if those samples are noisy or biased? Please provide sensitivity analysis across varying sample sets.
>
> To address the potential instability due to reliance on 10 unlabeled samples, we conducted five runs with different random seeds for 10, 50, and 100 samples, averaging performance on BoolQ: 82.93 ± 0.65% (10 samples), 82.57 ± 0.05% (50 samples), and 82.43 ± 0.02% (100 samples). The low variance across runs, particularly with larger sample sizes, demonstrates that our method is stable and minimally affected by sample variability.
>
> | Dataset | 10 Samples   | 50 Samples   | 100 Samples  |
> |---------|--------------|--------------|--------------|
> | BoolQ   | 82.93 ± 0.65 | 82.57 ± 0.05 | 82.43 ± 0.02 |
>
> > Scalability Concerns: The grid-search approach for layer/pruning percentage may not scale efficiently to larger models (e.g., 70B+ parameters).
>
> To address this, we extended our evaluation to the LLaMA3.1-70B-Instruct model, which has significantly more parameters (70B) than the 8B models. Our results below demonstrate that our method remains effective even for larger models, achieving improvements from 93.69% to 94.51% on SST2 and from 89.14% to 91.11% on BoolQ. While the grid search for optimal layer and pruning percentage (5% to 40% in 5% increments) incurs higher computational costs for larger models due to the increased number of neurons, the process is mitigated by our use of only 10 unlabeled samples for DSM neuron detection, keeping the overall computational overhead manageable. We will include additional results on more datasets and tasks in the next revision.
>
> Llama 3.1 70B Instruct:
>
> | Dataset | Base  | Ours      |
> |---------|-------|-----------|
> | SST2    | 93.69 | **94.51** |
> | BoolQ   | 89.14 | **91.11** |
>
> > What modifications would enable scaling the grid-search strategy to models with 100B+ parameters?
>
> Our method can be scaled to 100B+ models through the following lightweight modifications:
> 1. Parallelism: Grid evaluations are parallel and can be distributed across GPUs.
> 2. Coarse-to-Fine Search: A two-stage grid search-starting with coarse pruning percentages and refining only promising configurations-greatly reduces compute.
> 3. Layer Subsampling: Instead of searching all layers, we can pre-select a small subset based on IG statistics or transfer insights from smaller models (e.g., LLaMA 7B → 70B).

---

> > ### Author Response · Authors · 2025-06-01
> > **Response 2/2**
> >
> > > Narrow Task Scope: Evaluations are limited to multiple-choice tasks; generalization to generation tasks (e.g., summarization) or diverse domains (e.g., biomedical text) is untested.
> >
> > > Why restrict evaluations to multiple-choice tasks? Would the approach generalize to open-ended generation or domain-specific applications?
> >
> > Extending Integrated Gradients (IG) to open-ended generation tasks like summarization presents computational challenges, as IG is not inherently designed for autoregressive tasks, requiring attribution across sequential outputs, which can be expensive. However, this computation is a one-time cost during the DSM neuron detection phase, making it feasible with targeted optimizations. For instance, we can focus IG on key output tokens (e.g., summary-defining terms) or leverage task-specific adaptation sets to streamline attribution, enabling our pruning approach to generalize effectively to generation tasks. In this work, we prioritized multiple-choice tasks due to their structured output, which simplifies IG application by attributing contributions to the maximum logit, ensuring precise and interpretable DSM detection. The MMLU benchmark, covering 57 diverse subjects across STEM (e.g., mathematics, physics), humanities (e.g., history, philosophy), social sciences (e.g., economics), and professional fields (e.g., law, medicine), provides a robust foundation for assessing generalization across varied domains.  We are actively exploring adaptations for generation and specialized domains, and we will include a detailed discussion on these in the revised version.

---

### Comment · Area_Chair_YtKC · 2025-06-04
**Discussion period**

Dear reviewers,

This is a reminder that the discussion period is currently in progress and will end on June 10th.

I encourage you to read the other reviews as well as the responses by the authors and engage in a discussion.

Thanks a lot!

- AC

---

### Author Response · Authors · 2025-06-06
**Discussion Phase**

We thank all reviewers for their insightful feedback, which has strengthened our DSM neuron pruning method using Integrated Gradients for LLM generalization. Below, we summarize responses to key concerns, with details in the individual responses we posted earlier.

1. **Layer Selection (XM36, XAzb)**: Ablation studies show gate_proj pruning outperforms up_proj and down_proj (e.g., BoolQ: 82.35% vs. 81.84%, 79.53%), supporting its role in targeting DSMs. We will clarify this in the revised version of the paper.
2. **Sample Robustness (XM36, PXUN)**: Tests with 10 samples show low variance (e.g., BoolQ: 82.93 ± 0.65%) and match curated sets, suggesting stability. This will be included in the revised version of the paper.
3. **Task Scope (XM36, XAzb, u6ZY)**: Multiple-choice focus leverages structured outputs; MMLU’s diversity suggests broad applicability. We will discuss extending to generation tasks.
4. **Evaluation Clarity (u6ZY)**: We target in-domain generalization and will revise text to avoid cross-domain confusion.
5. **Computational Cost (u6ZY, PXUN)**: IG’s cost (3.74–4.68s/sample) is offset by using only 10 samples. Runtime comparisons will be added.

As the discussion period is nearing its end, we would appreciate your thoughts on whether these responses address your concerns.

---

### Decision · Program_Chairs · 2025-07-08

**Decision:**

Accept

**Comment:**

This paper proposes a new approach to adapt language models to specific downstream tasks. The core idea of the approach is to use integrated gradients (IG) to locate individual neurons that disproportionately influence high-confidence predictions of an (instruction fine-tuned) model, and subsequently prune these neurons to improve the generalization of the model to this particular task. The authors show promising results using as little as 10 samples to identify and prune neurons and find performance improvements across several multiple choice classification tasks.

Reviewers highlighted the conceptual novelty of the proposed approach, the practical importance for few-shot adaptation, as well as the extensive experiments.

The main weaknesses identified by the reviewers were a lack of a mechanistic justification for why the authors apply their method to a particular layer only, the somewhat narrow scope focusing only on classification tasks rather than generative ones, and a lack of experiments studying the variance of the proposed method.

Based on the author's responses, almost all of these concerns seem to have been addressed and the authors provided extensive explanations, elaborations, and new experimental results during the rebuttal.

A downside of the approach I'm seeing is that it is mostly applicable for tasks that have only a single output token such as classification. This is due to the reliance on IG which the authors also discussed during the rebuttal.

Overall, I agree with the reviewers that the proposed approach is interesting and provides an interesting perspective on how to perform task-adaptation for LLMs.